# TRIDENT PYRAMID NETWORKS: THE IMPORTANCE OF PROCESSING AT THE FEATURE PYRAMID LEVEL FOR BETTER OBJECT DETECTION

## ABSTRACT

Feature pyramids have become ubiquitous in multi-scale computer vision tasks such as object detection. Based on their importance, we divide a computer vision network into three parts: a backbone (generating a feature pyramid), a core (refining the feature pyramid) and a head (generating the final output). Most existing networks operating on feature pyramids, named cores, are shallow and mostly focus on communication-based processing in the form of top-down and bottom-up operations. We present a new core architecture called Trident Pyramid Network (TPN), that allows for a deeper design and for a better balance between communication-based processing and self-processing. We show consistent improvements when using our TPN core on the COCO object detection benchmark, outperforming the popular BiFPN baseline by 1.5 AP. Additionally, we empirically show that it is more beneficial to put additional computation into the TPN core, rather than into the backbone, by outperforming a ResNet-101+FPN baseline with our ResNet-50+TPN network by 1.7 AP, while operating under similar computation budgets. This emphasizes the importance of performing computation at the feature pyramid level in modern-day object detection systems. Code will be released.

## 1 INTRODUCTION

Many computer vision tasks such as object detection and instance segmentation require strong features both at low and high resolution to detect both large and small objects respectively. This is in contrast to the image classification task where low resolution features are sufficient as usually only a single object is present in the center of the image. Networks developed specifically for the image classification task (*e.g.* Simonyan & Zisserman (2014); He et al. (2016a); Xie et al. (2017)), further denoted by *backbones*, are therefore insufficient for multi-scale vision tasks. Especially poor performance is to be expected on small objects, as shown in Lin et al. (2017a).

In order to alleviate this problem, named the *feature fusion problem*, top-down mechanisms are added (Lin et al., 2017a) to propagate semantically strong information from the low resolution to the high resolution feature maps, with improved performance on small objects as a result. Additionally, bottom-up mechanisms can also be appended (Liu et al., 2018) such that the lower resolution maps can benefit from the freshly updated higher resolution maps. These top-down and bottom-up mechanisms can now be grouped into a layer, after which multiple of these layers can be concatenated, as done in Tan et al. (2020). We call this part of a computer vision network the *core*, laying in between the *backbone* and the task-specific *head* (see Figure 1). In general, we define a core module to be any module taking as input a feature pyramid and outputting an updated feature pyramid.

These top-down and bottom-up operations can be regarded as *communication*-based processing operating on two feature maps, as opposed to *content*-based self-processing operating on a single feature map. Existing cores such as FPN (Lin et al., 2017a), PANet (Liu et al., 2018) and BiFPN (Tan et al., 2020) mostly focus on communication-based processing, as this nicely supplements the backbone merely consisting of self-processing. However, when having multiple communication-based operations in a row, communication tends to saturate (everyone is up to date) and hence becomes superfluous. We argue it is therefore more effective to alternate communication-based processing

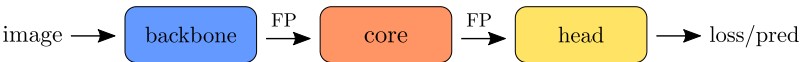

Figure 1: High-level view of a computer vision network. The backbone (*left*) processes the image to output a set of feature maps (*i.e.* a feature pyramid). The core (*middle*) takes in a feature pyramid (denoted by FP) and returns an updated feature pyramid. Finally, the head (*right*) produces the loss during training and makes predictions during inference from the final feature pyramid. In this work, we focus on improving the core.

with sufficient self-processing, such that feature maps have the time to come up with new findings to be communicated.

Based on this observation, we design the Trident Pyramid Network (TPN) core consisting of sequential top-down and bottom-up operations alternated with parallel self-processing mechanisms. The TPN core is equipped with hyperparameters controlling the amount of communication-based processing and self-processing. During the experiments, we empirically investigate what the optimal balance is between communication-based processing and self-processing (see Subsection 4.3).

The TPN core is compared to various baselines on the COCO object detection benchmark (Lin et al., 2014). Specific care is taken to ensure the baselines have similar computational characteristics, such that a fair comparison can be made. Using a ResNet-50 backbone and a simple one-stage detector head, our TPN core peaks at 41.8 AP on the COCO validation set when using the 3x training schedule (see Subsection 4.2). This is a 1.5 AP improvement over a BiFPN core of similar computational expense.

When having additional compute to improve performance, practitioners typically decide to replace their backbone with a heavier one. A ResNet-50+FPN network for example gets traded for the heavier ResNet-101+FPN network. Yet, one might wonder whether it is not more beneficial to add additional computation into the core (*i.e.* at the feature pyramid level) by using a ResNet-50+TPN network, rather than into the backbone by using a ResNet-101+FPN network. When comparing both options under similar computational characteristics, we show a 1.7 AP improvement of the ResNet-50+TPN network over the ResNet-101+FPN network. This empirically shows that it is more beneficial to add additional computation into the core, highlighting the importance of performing computation at the feature pyramid level in modern-day object detection systems. We hope this new insight drives researchers to design even better cores in the future.

## 2 RELATED WORK

In order to obtain multi-scale features, early detectors performed predictions on feature maps directly coming from the backbone, such as MS-CNN (Cai et al., 2016) and SSD (Liu et al., 2016). As the higher resolution maps from the backbone contain relatively weak semantic information, top-down mechanisms were added to propagate semantically strong information from lower resolution maps back to the higher resolution maps as in FPN (Lin et al., 2017a) and TDM (Shrivastava et al., 2016). Since, many variants and additions have been proposed: PANet (Liu et al., 2018) appends bottom-up connections, M2det (Zhao et al., 2019) uses a U-shape feature interaction architecture, ZigZagNet (Lin et al., 2019) adds additional pathways between different levels of the top-down and bottom-up hierarchies, NAS-FPN (Ghiasi et al., 2019) and Hit-Detector (Guo et al., 2020) use Neural Architecture Search (NAS) to automatically design a feature interaction topology, and BiFPN (Tan et al., 2020) modifies PANet by removing some connections, adding skip connections and using weighted feature map aggregation. All of the above variants focus on improving the communication between the different feature maps. We argue however that to be effective, extra content-based self-processing is needed in between the communication flow.

Not all methods use a feature pyramid to deal with scale variation. TridentNet (Li et al., 2019) applies parallel branches of convolutional blocks with different dilations on a single feature map to obtain scale-aware features. In DetectoRS (Qiao et al., 2021), they combine this idea with feature pyramids, by applying their switchable atrous convolutions (SAC) inside their recursive feature pyramids (RFP). Note that to avoid any name confusion with TridentNet, we call our core by its abbreviated name TPN as opposed to Trident Pyramid Network.

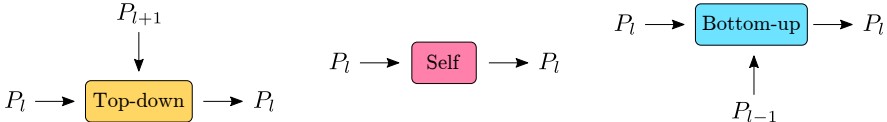

Figure 2: Collection of building blocks for core architecture design. Here $P_l$ denotes feature map of level $l$ which is $2^l$ times smaller compared to the initial image resolution. (*Left*) General top-down operation updating feature map $P_l$ with information from lower resolution map $P_{l+1}$. (*Middle*) General self-processing operation updating feature map $P_l$ with information from itself, *i.e.* from feature map $P_l$. (*Right*) General bottom-up operation updating feature map $P_l$ with information from higher resolution map $P_{l-1}$.

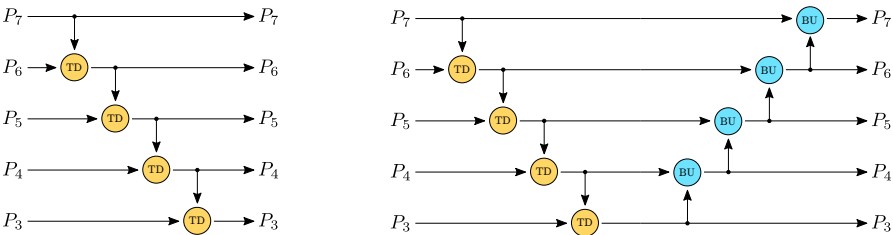

Figure 3: (*Left*) A layer from the FPN core architecture. (*Right*) A layer from the PANet core architecture.

Our TPN core is also related to networks typically used in segmentation such as U-Net (Ronneberger et al., 2015) and stacked hourglass networks (Newell et al., 2016), given that these networks also use a combination of top-down, self-processing and bottom-up operations. A major difference of these networks with our TPN core however, is that they do not operate on a feature pyramid in the sense that lower resolution maps are only generated and used within a single layer (*e.g.* within a single hourglass) and are not shared across layers (*e.g.* across two neighboring hourglasses).

Finally, note that some works such as Guo et al. (2020) and Bochkovskiy et al. (2020) refer to the the network part connecting the backbone with the head as the neck (instead of the core). That name implies that the neck is merely a connection piece between the backbone and the head, and is of little importance. Yet, we show that the neck is in fact an important part of the network, and therefore call it the core instead.

## 3 METHOD

### 3.1 TPN CORE ARCHITECTURE

Generally speaking, the core receives a feature pyramid as input, and outputs an updated feature pyramid. Here, a feature pyramid is defined as a collection of feature maps, with feature maps defined as a collection of feature vectors (called features) organized in a two-dimensional map. More specifically, feature map $P_l$ denotes a feature map of level $l$ which is $2^l$ times smaller in width and height compared to the initial image resolution. A popular choice for the feature pyramid (Lin et al., 2017b) is to consider feature maps $\{P_3, P_4, P_5, P_6, P_7\}$, which we will use as the default setting throughout our discussions and experiments.

The core is constructed from three building blocks: top-down operations, self-processing operations and bottom-up operations (see Figure 2). In this subsection, we focus on how these operations are best combined, independently of their precise implementations. We call this configuration of operations making up a core, the *core architecture*. The specific implementations corresponding to the top-down, self-processing and bottom-up operations will be discussed in Subsection 3.2 and Subsection 3.3.

Using these general building blocks, we can recreate the popular FPN (Lin et al., 2017a) and PANet (Liu et al., 2018) core architectures in Figure 3. Note that the architectures slightly differ from those found in the original works (Lin et al., 2017a; Liu et al., 2018), as the input layers are missing. Given

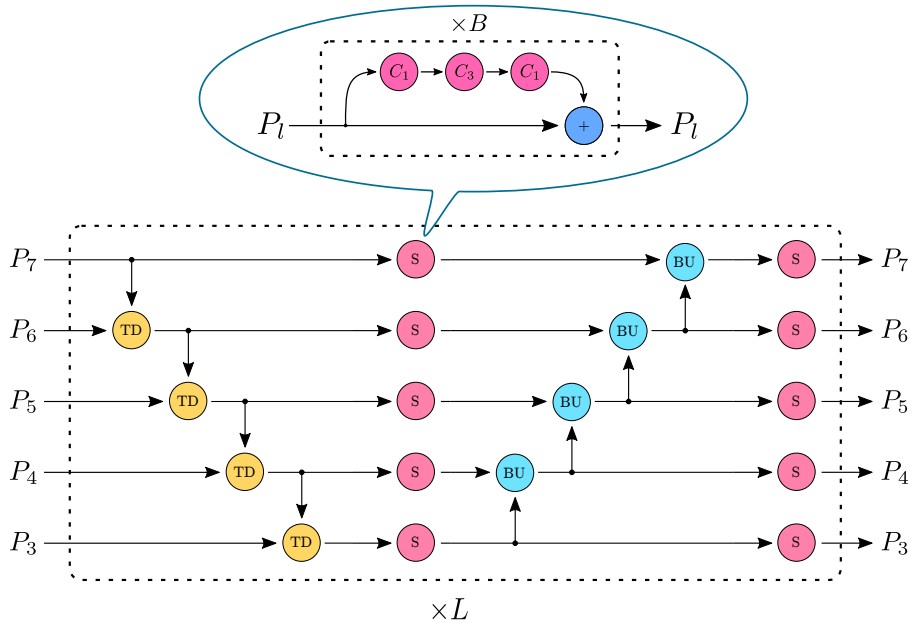

Figure 4: Our TPN core architecture consisting of $L$ consecutive TPN core layers (*bottom*), with each self-processing operation consisting of $B$ consecutive bottleneck layers (*top*).

that these input layers are meant to transition from backbone feature sizes to core feature sizes, we decided to move these transition layers from the core to the backbone instead, such that multiple core layers can easily be concatenated. Note moreover that Figure 3 only defines the architecture of the core, without specifying the implementation of the top-down and bottom-up operations. These implementations could hence differ from those found in the original works (Lin et al., 2017a; Liu et al., 2018).

From the FPN and PANet core architectures from Figure 3, we make following two observations. First, we can see that the top-down and bottom-up operations are sequential. Secondly, we observe the lack of self-processing operations in both core architectures. In what follows, we discuss both aspects in more depth.

First, we discuss the trade-off between sequential and parallel operations in greater detail. By *sequential* operations, we mean that $P_l$ is updated with the new $P_{l\pm1}$ instead of with the old one, forcing the operations to be performed sequentially as the new feature maps must be available. Alternatively, one could instead opt for *parallel* operations by solely relying on the old feature maps. The choice between parallel and sequential could be regarded as a trade-off between speed and accuracy, while maintaining a similar memory consumption. Given that the top-down and bottom-up operations can be quite expensive, especially on high resolution maps, it is important to get the most out of every single operation. We hence believe that the sequential variant should be preferred here for the top-down and bottom-up operations, as found in the FPN and PANet core architectures.

Secondly, we discuss the lack of self-processing operations in the FPN and PANet core architectures. When looking at the PANet architecture, we see that bottom-up operations immediately follow the top-down operations. We argue that this is sub-optimal. Take for example a look at the $P_4$-$P_3$ top-down operation, followed immediately by the $P_3$-$P_4$ bottom-up operation. The $P_3$ map was just updated with information from $P_4$ and now $P_3$ must immediately communicate its content back to $P_4$ before having the possibility to digest and work on this new information. We hence argue that the top-down and bottom-up operations should be separated with self-processing operations. This gives the feature maps the opportunity to work on themselves before communicating back to their peers.

By combining the insights from previous two discussions, we arrive at the Trident Pyramid Network (TPN) core architecture consisting of sequential top-down and bottom-up operations alternated with self-processing operations (see lower part of Figure 4). The name is inspired by the top-down, first

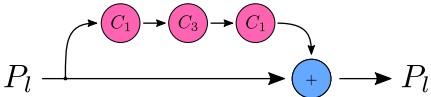

Figure 5: Bottleneck layer used as base self-processing operation. It is a skip-connection operation with a residual branch consisting of three convolution operations: a convolution operation of kernel size 1 reducing the original feature size to the hidden feature size, a content convolution operation of kernel size 3 applied on the hidden feature size, and finally a convolution operation of kernel size 1 expanding the hidden feature size back to the original feature size. Note that each convolution operation (*i.e.* pink convolution node) consists of the actual convolution preceded (He et al., 2016b) by group normalization (Wu & He, 2018) and a ReLU activation function.

self-processing and bottom-up operations resembling a trident. Note that the TPN self-processing operations happen in parallel. This is not necessary, but we believe this to be the most natural choice.

## 3.2 COMMUNICATION VS. CONTENT

When looking at the TPN core in Figure 4, we have on the one hand communication-based top-down and bottom-up operations, and on the other hand content-based self-processing operations. One might now wonder what the optimal balance between communication and content is. In order to investigate this matter, we introduce hyperparameters enabling us to control the amount of content-based processing on the one hand, and the amount of communication-based processing on the other hand.

First, we take a closer look at the self-processing operation. In general, we consider this self-processing operation to be a sequence of layers, with each layer containing the base self-processing operation. In this paper, we chose the bottleneck architecture from He et al. (2016a) (see Figure 5) as the base self-processing operation. The number of bottleneck layers $B$ per self-processing operation then determines the amount of content-based processing within a TPN layer.

Secondly, with each TPN layer consisting of one top-down and one bottom-up sequence, the number of TPN core layers $L$ determines the amount of communication-processing within the TPN core. Note that while the total amount of communication-based processing only depends on the number of TPN layers $L$, the total amount of self-processing within TPN depends on both the number of bottleneck layers per self-processing operation $B$, as well as on the total number of TPN layers $L$. In Figure 4, an overview of our TPN core architecture is shown, displaying the number of bottleneck layers $B$ and the number of TPN core layers $L$. By varying these hyperparameters $B$ and $L$, we can hence control the balance between content-based processing and communication-based processing respectively. In Subsection 4.3, we empirically find out which combinations work best.

## 3.3 TOP-DOWN AND BOTTOM-UP OPERATIONS

Let us now take a closer look at the top-down and bottom-up operations. Generally speaking, these operations update a feature map based on a second feature map, either having a lower resolution (top-down case) or a higher resolution (bottom-up case). Our implementation of the top-down and bottom-up operations are shown in Figure 6. The operations consist of adding a modified version of $P_{l\pm1}$ to $P_l$. This is similar to traditional skip-connection operations, with the exception that the *residual features* originate from a different feature map. The residual branch of the top-down operation consists of a linear projection followed by bilinear interpolation. The presence of the linear projection is important here, as it makes the expectation of the residual features zero at initialization. Failing to do so can be detrimental, especially when building deeper core modules, as correlated features add up without constraints. An alternative consists in replacing the blue addition nodes with averaging nodes. This however fails to keep the skip connection computation free (due to the 0.5 factor), which is undesired (He et al., 2016b). The residual branch of the bottom-up operation is similar to the bottleneck residual branch in Figure 5. Only the middle $3 \times 3$ convolution has stride 2 instead of stride 1, avoiding the need for an interpolation step later in the residual branch.

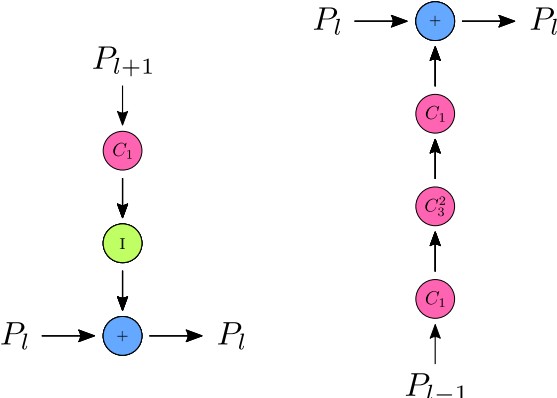

Figure 6: Implementation of the top-down (*left*) and bottom-up (*right*) operations. The pink convolution nodes are defined as in Figure 5, with the subscript denoting the kernel size and the superscript denoting the stride (stride 1 when omitted). The green node is an interpolation node resizing the input feature map to the required resolution by using bilinear interpolation.

## 4 EXPERIMENTS

### 4.1 SETUP

**Dataset.** We perform our experiments on the 2017 COCO dataset (Lin et al., 2014), where we train on the 118k training images and evaluate on the remaining 5k validation images.

**Implementation details.** Throughout our experiments, we use an ImageNet (Deng et al., 2009) pretrained ResNet-50 (or ResNet-101) backbone (He et al., 2016a), with frozen stem, stage 1 and batchnorm layers (see Radosavovic et al. (2020) for used terminology).

Our feature pyramid consists of five feature maps, ranging from $P_3$ to $P_7$, each having feature size 256. Our initial feature pyramid is constructed based on the backbone output feature maps $C_3$ to $C_5$ from stages 2, 3 and 4 respectively. Remember that the subscript denotes how many times the feature map was downsampled with factor 2 compared to the input image. The initial $P_3$ to $P_5$ maps are obtained by applying simple linear projections on $C_3$ to $C_5$, whereas the initial $P_6$ and $P_7$ maps are obtained by applying a simple network on $C_5$ consisting of 2 convolutions with stride 2, with a ReLU activation in between (similar to Lin et al. (2017b)). Throughout our TPN core modules, we use group normalization (Wu & He, 2018) with 8 groups. For the bottleneck layers (see Figure 5), we use a hidden feature size of 64.

As detection head, we use the one-stage detector head from RetinaNet (Lin et al., 2017b), with 1 or 4 hidden layers in both classification and bounding box subnets. We follow the implementation and settings from Wu et al. (2019), except that the last layer of the subnets has kernel size 1 (instead of 3) and that we normalize the losses per feature map (instead of over the whole feature pyramid).

We train our models with the AdamW optimizer (Loshchilov & Hutter, 2017) with weight decay $10^{-4}$ using an initial learning rate of $10^{-5}$ for the backbone parameters and an initial learning rate of $10^{-4}$ for the remaining model parameters. Our main experiment results in Subsection 4.2 are obtained by using the 3x training schedule, consisting of 36 epochs with learning rate drops after the 27th and 33rd epoch with a factor 0.1. Our models are trained on 2 GPUs with batch size 2, while using the same data augmentation scheme as in Carion et al. (2020).

### 4.2 MAIN TPN EXPERIMENTS

**Baselines.** In this subsection, we perform experiments to evaluate the TPN core. As baseline, we consider the BiFPN core architecture from Tan et al. (2020), with Swish-1 activation functions replaced by ReLU activation functions. Multiple of these BiFPN layers will be concatenated such that the BiFPN core shares similar computational characteristics compared to the tested TPN modules.

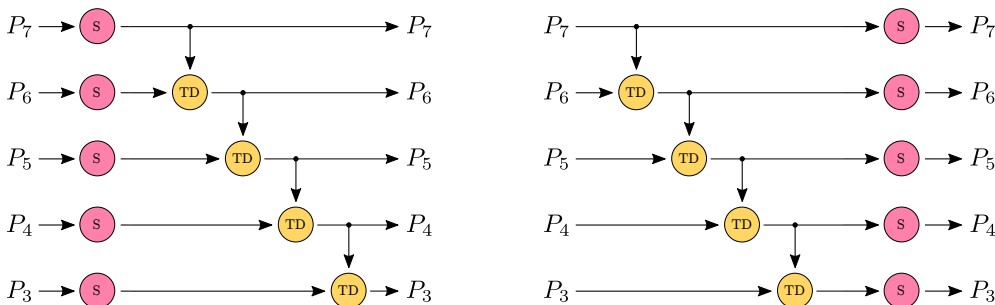

Figure 7: (*Left*) The baseline bFPN core architecture simulating a heavier backbone followed by a single FPN layer. (*Right*) The baseline hFPN core architecture simulating a single FPN layer followed by a heavier head.

Table 1: Experiment results on the 2017 COCO validation set of different TPN cores (top four rows) and its baselines (bottom five rows). The five leftmost columns specify the network (Back = Backbone, Core, $B$ = Number of bottleneck layers per self-processing node, $L$ = Number of layers, $C$ = Number of hidden layers in classification and bounding box subnets), the middle six columns show its performance and the five rightmost columns show its computational characteristics. These characteristics were obtained on a GeForce GTX 1660 Ti GPU by applying the network on a batch of two $800 \times 800$ images, each containing 10 ground-truth objects during training. The training characteristics are found under the columns 'Params', 'tFPS' and 'tMem', whereas the inference characteristics are found under the 'iFPS' and 'iMem' columns. Here the FPS metric should be interpreted as the number of times the GPU can process above input. Note that both forward and backward passes (with parameter update from the optimizer) are used to obtain the training characteristics.

| Back | Core | $B$ | $L$ | $C$ | AP | $AP_{50}$ | $AP_{75}$ | $AP_S$ | $AP_M$ | $AP_L$ | Params | tFPS | tMem | iFPS | iMem |
|------|------|-----|-----|-----|------|------|------|------|------|------|--------|------|--------|------|---------|
| R50 | TPN | 7 | 1 | 1 | 41.3 | 60.5 | 44.2 | 26.3 | 45.9 | 52.5 | 36.3 M | 1.7 | 3.31 GB | 5.3 | 0.50 GB |
| R50 | TPN | 3 | 2 | 1 | 41.6 | 60.9 | 44.6 | **26.4** | 45.8 | 53.2 | 36.2 M | 1.7 | 3.21 GB | 5.5 | 0.50 GB |
| R50 | TPN | 2 | 3 | 1 | **41.8** | 61.1 | 44.4 | 26.2 | 46.1 | **53.7** | 36.7 M | 1.6 | 3.27 GB | 5.3 | 0.50 GB |
| R50 | TPN | 1 | 5 | 1 | **41.8** | **61.2** | **45.0** | 26.0 | **46.3** | 53.4 | 37.1 M | 1.6 | 3.22 GB | 5.3 | 0.50 GB |
| R50 | BiFPN | − | 7 | 1 | 40.3 | 59.8 | 43.3 | 24.5 | 44.1 | 52.2 | 34.7 M | 1.9 | 3.28 GB | 6.1 | 0.49 GB |
| R50 | bFPN | 14 | − | 1 | 39.6 | 60.3 | 42.4 | 24.2 | 43.5 | 51.3 | 36.1 M | 1.7 | 3.26 GB | 5.4 | 0.49 GB |
| R50 | hFPN | 14 | − | 1 | 40.0 | 60.2 | 43.0 | 25.6 | 43.9 | 51.1 | 36.1 M | 1.7 | 3.26 GB | 5.4 | 0.49 GB |
| R101 | FPN | − | 1 | 4 | 40.1 | 60.1 | 42.8 | 24.0 | 44.0 | 52.7 | 55.1 M | 1.4 | 3.20 GB | 4.1 | 0.57 GB |
| R101 | TPN | 2 | 1 | 1 | 40.9 | 61.0 | 44.2 | 25.0 | 45.3 | 52.6 | 51.7 M | 1.6 | 3.20 GB | 4.6 | 0.55 GB |

As the FPN layer was not designed to be concatenated many times, we provide two additional baselines using a single FPN (see Figure 7). Here the bFPN baseline performs additional self-processing before the FPN simulating a heavier backbone, while the hFPN baseline performs additional self-processing after the FPN simulating a heavier head. As such, we will not only be able to evaluate whether the TPN core outperforms other cores, but also whether it outperforms detection networks using a simple core with heavier backbones or heads, while operating under similar computation budgets.

Finally, we also compare ResNet-101+FPN and ResNet-101+TPN networks with a ResNet-50+TPN network of similar computation budget, to further assess whether it is more beneficial to put additional computation into the backbone or into the core.

**Results.** The experiment results evaluating four different TPN configurations against the five baselines, are found in Table 1.

First, notice how the $B$ and $L$ hyperparameters (defining the TPN configuration) are chosen in order to obtain models with similar computational characteristics. Here hyperparameter $B$ denotes the number of bottleneck layers per self-processing node, while hyperparameter $L$ denotes the number of consecutive core layers (see Figure 4). These similar computational characteristics ensure us that a fair comparison can be made between the different models.

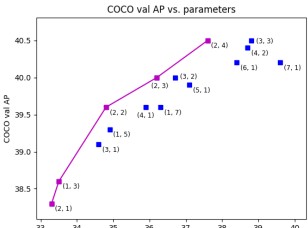 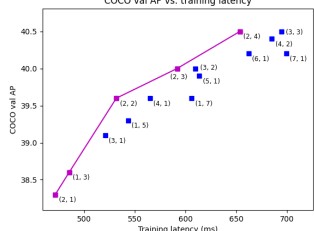 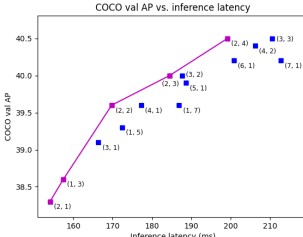

Figure 8: Accuracy vs. efficiency comparisons between 15 different $(L, B)$ TPN configurations using the 'parameters' (*left*), 'training latency' (*middle*) and 'inference latency' (*right*) efficiency metrics. The accuracies correspond to the COCO validation APs, obtained after training the models for 12 epochs using the 1x schedule. The TPN configurations yielding the best accuracy vs. efficiency trade-off at various computation budgets, are highlighted in magenta.

Secondly, we observe that the results between the four different TPN configurations are very similar, all four obtaining between $41.3$ and $41.8$ AP. At first glance, it appears that having more TPN core layers $L$ is slightly more beneficial than having more bottleneck layers $B$ under similar computational budgets. In Subsection 4.3, we will further investigate which TPN configurations yield the best accuracy vs. efficiency trade-off at various computation budgets.

Thirdly, when comparing our TPN cores (top four rows) with the BiFPN core (fifth row), we clearly see the superiority of the TPN core. All four TPN configurations outperform the BiFPN core with $1.0$ up to $1.5$ AP, where improvements are found across all object sizes. Note that the BiFPN baseline has slightly fewer parameters and is slightly faster compared to the used TPN cores. However, when using additional BiFPN layers to better align the BiFPN computation budget with the TPN budgets, we did not obtain any performance improvements. We therefore chose to report the results of a slightly lighter BiFPN core instead. In Appendix A.1, we provide additional results where we compare the TPN core with the BiFPN core when using a large backbone.

Fourthly, when comparing the ResNet-50+TPN networks (top four rows) with the ResNet-50+bFPN, ResNet-50+hFPN, ResNet-101+FPN and ResNet-101+TPN baselines (bottom four rows), we again see that the ResNet-50+TPN networks work best. The best-performing baseline (ResNet-101+TPN) from this category is outperformed by all four ResNet-50+TPN configurations with $0.4$ up to $0.9$ AP. Note that the ResNet-101+TPN baseline has considerably more parameters and is clearly slower at inference, but still does not match the performance of the ResNet-50+TPN networks despite its higher computational cost. This hence shows that the TPN core not only outperforms other cores such as BiFPN, but also other detection networks using heavier backbones or heads while operating under similar overall computation budgets. This highlights the importance of core modules operating on feature pyramids in general object detection networks.

### 4.3 COMPARISON BETWEEN DIFFERENT TPN CONFIGURATIONS

In this subsection, we investigate which TPN configurations yield the best accuracy vs. efficiency trade-off at various computation budgets. Here, a TPN configuration is determined by the hyperparameter pair $(L, B)$, respectively denoting the number of TPN layers and the number of bottleneck layers per self-processing node. We use the same settings as explained in Subsection 4.1, except that we only train for 12 epochs using the 1x training schedule. In Figure 8, we compare 15 different $(L, B)$ TPN configurations using the 'parameters', 'training latency' and 'inference latency' efficiency metrics, with the latency metrics obtained using the same methodology as in Table 1.

We can see from the magenta curves yielding the optimal TPN configurations at various computation budgets, that having a good balance between communication-based processing (in the form of TPN layers $L$) and self-processing (in the form of bottleneck layers per self-processing node $B$) is important. We can for example see that the balanced $(2, 2)$ configuration outperforms the unbalanced $(3, 1)$ and $(1, 5)$ configurations. The same observation can be also made at higher computation budgets, where the balanced $(2, 4)$, $(4, 2)$ and $(3, 3)$ configurations outperform the unbalanced $(6, 1)$ and $(7, 1)$ configurations.

Table 2: Comparison of our best-performing TPN model with other prominent object detection networks on the 2017 COCO validation set (see Table 1 for the definitions of $B$, $L$ and $C$). All models use a ResNet-50 (He et al., 2016a) backbone. The number of FLOPS are computed as in Carion et al. (2020), by applying the `flop_count_operators` tool from Detectron2 (Wu et al., 2019) on the first 100 images of the validation set. The number of inference FPS 'iFPS' is calculated as explained in Table 1, while using the implementations provided by MMDetection (Chen et al., 2019) for the baseline models.

| Model | $B$ | $L$ | $C$ | Epochs | AP | $AP_{50}$ | $AP_{75}$ | $AP_S$ | $AP_M$ | $AP_L$ | Params | GFLOPS | iFPS |
|---|---|---|---|---|---|---|---|---|---|---|---|---|---|
| Faster R-CNN+FPN (Wu et al., 2019) | – | 1 | 4 | 37 | 40.2 | 61.0 | 43.8 | 24.2 | 43.5 | 52.0 | 41.5 M | 180 | 4.1 |
| RetinaNet+FPN (Wu et al., 2019) | – | 1 | 4 | 37 | 38.7 | 58.0 | 41.5 | 23.3 | 42.3 | 50.3 | 37.7 M | 208 | 5.1 |
| DETR (Carion et al., 2020) | – | – | – | 500 | 42.0 | 62.4 | 44.2 | 20.5 | 45.8 | 61.1 | 41.3 M | 86 | 7.3 |
| Deformable DETR (Zhu et al., 2020) | – | – | – | 50 | 43.8 | 62.6 | 47.7 | 26.4 | 47.1 | 58.0 | 39.8 M | 173 | 3.5 |
| RetinaNet+TPN (**ours**) | 2 | 3 | 1 | 36 | 41.8 | 61.1 | 44.4 | 26.2 | 46.1 | 53.7 | 36.7 M | 121 | 5.3 |

We hence empirically show that balanced $(L, B)$ configurations with $L \geq 2$ and $B \geq 2$ are to be preferred over unbalanced configurations such as $(L, 1)$ and $(1, B)$. By having only one self-processing operation in between each communication-based operation, existing cores with a $(L, 1)$ configuration such as PANet (Liu et al., 2018) and BiFPN (Tan et al., 2020) crucially lack self-processing. The TPN core solves this problem by introducing the $(L, B)$ hyperparameter pair, such that a better balance between communication-based processing and self-processing can be chosen.

### 4.4 COMPARISON WITH OTHER OBJECT DETECTION NETWORKS

In Table 2, we compare the performance of our best-performing model against other prominent object detection networks. We make following two sets of observations.

First, we compare our model with the popular two-stage object detector Faster R-CNN+FPN (Ren et al., 2015; Lin et al., 2017a) and the popular one-stage object detector RetinaNet+FPN (Lin et al., 2017b;a) (top two rows). Under similar training schedules, we observe that our model performs better across all object sizes, while being computationally cheaper (three rightmost columns). Note that the RetinaNet+FPN model closely resembles our RetinaNet+TPN model, except that they use a FPN core and a heavier head using 4 hidden layers in the classification and bounding box subnets (Wu et al., 2019) instead of 1. The results again show that it is more beneficial to put additional computation into the core (RetinaNet+TPN) rather than into the head (RetinaNet+FPN).

Secondly, we compare our model with the DETR (Carion et al., 2020) and Deformable DETR (Zhu et al., 2020) models (middle two rows). These are two recent object detectors based on transformers with set prediction. On small objects, we observe similar results for our model and Deformable DETR, while DETR performs significantly worse. This can be understood from the fact that DETR only operates on a single low resolution feature map instead of on a feature pyramid, once again proving the importance of feature pyramids. On large objects on the other hand, we see a clear superiority of the transformer-based detectors with set prediction compared to our model using a simple RetinaNet-like head. Given that our TPN model only specifies the core module, we could combine it with this new type of detection head that uses a transformer decoder combined with set prediction (Carion et al., 2020). We believe this should improve the performance of our TPN model for large objects in a similar way as for the DETR and Deformable DETR models. We will further investigate this in future work.

## 5 CONCLUSION

In this paper, we introduced a new type of core architecture, called the Trident Pyramid Network (TPN). We show consistent improvements when using our TPN core on the COCO object detection benchmark compared to various baselines. From our TPN experiments, we see that both communication-based processing and self-processing are crucial within core modules in order to obtain good results. We additionally observe that our TPN performance could not be matched by heavier backbones or heads under similar overall computation budgets, showing the importance and effectiveness of the core within object detection systems.

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

# A  APPENDIX

## A.1  COMPARISON BETWEEN TPN AND BiFPN WHEN USING LARGE BACKBONE

In this appendix subsection, we provide additional results comparing the TPN core with the BiFPN core, when using the large ResNeXt-101-32x4-DCNv2 (Xie et al., 2017; Zhu et al., 2020) backbone. To carry out these experiments, we chose the TPN core with $L = 3$ TPN layers, and $B = 2$ bottleneck layers per self-processing node. We follow the same settings as explained in Subsection 4.1, except that we use the 1x training schedule and train on only one GPU instead of two.

The results are found in Table 3. We can see that the TPN core outperforms the BiFPN core by $0.7$ AP, while having a similar computation budget. We can moreover see that the TPN core has a better performance across all object scales. These additional experimental results show that the superiority of the TPN core compared to the BiFPN core generalizes to larger backbones.

Table 3: Experiment results on the 2017 COCO validation set comparing the TPN core (top row) with the BiFPN core (bottom row), when using the large ResNeXt-101-32x4-DCNv2 backbone. The six leftmost columns specify the model and training settings, the six middle columns show the model performance, and the five rightmost columns contain the computational characteristics of the model. These characteristics are obtained as explained in Table 1, except that we use a batch of two $600 \times 600$ images (instead of two $800 \times 800$ images) to avoid an out-of-memory (OOM) error on the used GeForce GTX 1660 Ti GPU.

| Backbone | Core | $B$ | $L$ | $C$ | Epochs | AP | $AP_{50}$ | $AP_{75}$ | $AP_S$ | $AP_M$ | $AP_L$ | Params | tFPS | tMem | iFPS | iMem |
|---|---|---|---|---|---|---|---|---|---|---|---|---|---|---|---|---|
| X101-DCNv2 | TPN | 2 | 3 | 1 | 12 | 44.1 | 64.9 | 47.7 | 27.9 | 48.6 | 58.0 | 59.1 M | 1.1 | 3.60 GB | 4.7 | 0.39 GB |
| X101-DCNv2 | BiFPN | – | 7 | 1 | 12 | 43.4 | 64.1 | 46.9 | 26.6 | 47.9 | 56.7 | 57.1 M | 1.2 | 3.59 GB | 5.1 | 0.38 GB |

