# OpenReview forum: "Trident Pyramid Networks: The importance of processing at the feature pyramid level for better object detection"
_ICLR.cc/2022/Conference — ICLR 2022 Submitted_

### Official Review · Reviewer_MSxk · 2021-10-31

**Correctness:** 4
**Technical Novelty And Significance:** 2
**Empirical Novelty And Significance:** 2
**Recommendation:** 5
**Confidence:** 5

**Main Review:**

The paper is very well written. The experiments demonstrate TPN brings solid improvement (1.5% on COCO) over existing works on RetinaNet w/ ResNet50.

The technical novelty of this paper is limited. Stacking FPN is not new as BiFPN and PANet also stacks multiple FPN to gain better performance which is also mentioned by the authors. So the major difference between TPN and existing works is that TPN replaces the plain skip connections within each FPN with residual modules. It is interesting to see that replacing skip connections with residual module is a more effective way to improve performance than stacking FPNs. But this change is incremental, and I do not consider this to be a novel design.

TPN introduces two hyper-parameters to control the number of stacks and the number of residual modules. Experiments in table 1 show that how the performance change with the two new hyper-parameters under a fixed computational budget. It would be interesting to see how the performance would change with the computational budgets and different hyper-parameter choices. The authors can consider an efficiency and accuracy tradeoff curve, which would give a better understanding on the importance of the residual modules.

The authors should try their approach on a deeper network such as ResNet101. It is unclear if the proposed approach would generalize to deeper networks. The authors should also include FPS in Table 2 when they compare TPN with other prominent models. There are other factors that can impact the inference speed of a model such as memory latency and whether the network is parallelizable other than GFLOPS.

**Summary Of The Paper:**

This paper proposes a new feature pyramid network, trident pyramid network (TPN). TPN stacks multiple FPNs together and a FPN processes the features produced by its previous stage. TPN also replaces the skip connections within each FPN with residual modules to further process the features. Experiments show that TPN outperforms existing works such as BiFPN and PANet on ResNet50 under a similar computational budget by varying the number of FPNs and residual modules.

**Summary Of The Review:**

Although TPN shows some interesting empirical results and solid improvement over existing works on smaller networks, it is not technically interesting, and the novelty is limited.

---

> ### Author Response · Authors · 2021-11-18
> **Reply to R4**
>
> > **R4.1.1**: *The technical novelty of this paper is limited. Stacking FPN is not new as BiFPN and PANet also stacks multiple FPN to gain better performance which is also mentioned by the authors.*
>
> **A4.1.1**: See our general comment.
> ***
>
> > **R4.1.2**: *So the major difference between TPN and existing works is that TPN replaces the plain skip connections within each FPN with residual modules. It is interesting to see that replacing skip connections with residual module is a more effective way to improve performance than stacking FPNs. But this change is incremental, and I do not consider this to be a novel design.*
>
> **A4.1.2**: See A2.1.2.
> ***
>
> > **R4.2**: *It would be interesting to see how the performance would change with the computational budgets and different hyper-parameter choices. The authors can consider an efficiency and accuracy tradeoff curve, which would give a better understanding on the importance of the residual modules.*
>
> **A4.2**: Following your suggestion, we trained $15$ different $(L, B)$ TPN configurations of various computation budgets for $12$ epochs using the 1x training schedule. The results can be found in Figure 8 of the revised paper. We conclude from the experiments that balanced $(L, B)$ configurations with $L\geq2$ and $B\geq2$ give a better accuracy vs. efficiency trade-off compared to unbalanced $(L, 1)$ and $(1, B)$ configurations. A balanced $(L, B)$ configuration with fewer TPN layers $L$ but more bottleneck layers $B$ (i.e. residual modules), hence outperforms a $(L, 1)$ configuration of similar computation budget, highlighting the importance of self-processing in Pyramid Networks (PNs). See Subsection 4.3 of the revised paper for a more extensive discussion. Thanks for your great suggestion!
> ***
>
> > **R4.3.1**: *The authors should try their approach on a deeper network such as ResNet101. It is unclear if the proposed approach would generalize to deeper networks.*
>
> **A4.3.1**: See R2.4 where we compare a ResNeXt-101-DCNv2+TPN model with a ResNeXt-101-DCNv2+BiFPN model.
> ***
>
> > **R4.3.2**: *The authors should also include FPS in Table 2 when they compare TPN with other prominent models. There are other factors that can impact the inference speed of a model such as memory latency and whether the network is parallelizable other than GFLOPS.*
>
> **A4.3.2**: We will include FPS in Table 2 in our revision. Thanks for your suggestion.

---

> ### Comment · Reviewer_MSxk · 2021-11-29
> **Limited technical novelty**
>
> I appreciate the authors' efforts on addressing most of our concerns by providing detail explanations and new experiments. But I am not going to adjust my score for this paper because I am still concerned about the novelty of this paper. The authors claim that their main technical novelty lies in what is being stacked in TPN (stacking fewer TPN layers and replacing plain connections with stacked residual modules). I do not consider this to be significant because these changes are trivial and it only slightly outperforms existing works on larger networks. Having said that, this paper does show some interesting empirical results on how we can a better TPN. Although I will not recommend accepting this paper, I am fine if it is accepted.

---

### Official Review · Reviewer_YsZ7 · 2021-11-01

**Correctness:** 3
**Technical Novelty And Significance:** 3
**Empirical Novelty And Significance:** 3
**Recommendation:** 6
**Confidence:** 4

**Main Review:**

This work proposes a new core architecture (called the Trident Pyramid Network) for object detection CNNs and claims that:
1. A heavier core enhances object detection performance.
2. It is better to invest any extra computational budget on a heavier core rather than on a heavier backbone.

Experimental studies that compare the proposed method with counterparts like PANet, FPN, and Faster-RCNN+FPN on the COCO object detection benchmark corroborate the above claim 1. However, empirical evidence for claim 2 is not convincing enough. The quality of the paper may be enhanced if some concerns (see cons below) are addressed during the rebuttal period.

--------------------------
Pros:
--------------------------
1. The paper is well-written. The flow of the paper is engaging. The method is well-motivated and the literature review is sufficient.

2. Figures are informative. For example, Figure 4 helps the reader visualize the architecture of the  proposed TPN core.

3. The trade-off between computational cost and detection performance of the proposed TPN can be controlled by two hyperparameters, namely B (controlling the amount of self-processing) and L (controlling the amount of communication-based processing). This enables the user to choose the necessary amount of self-processing versus communication-based processing.

4. Experimental studies with various B and L values for TPN seem to outperform other methods under comparison. The proposed method’s performance is shown to be 3 AP and 1.5 AP better than PANet and BiFPN respectively (with the same backbone network and under similar computational cost). In addition, TPN outperformed RetinaNet and Faster-RCNN+FPN by 3.1 AP and 1.6 AP respectively,  convincing the reader that the proposed method is effective.

5. The proposed method is shown to outperform other CNN based  architectures like Faster-RCNN+FPN and RetinaNet.

6. Although TPN underperforms compared to vision transformer based architectures like, DETR and deformable DETR, the paper claims to adopt transformer based processing in the TPN core to enhance performance in future work.


--------------------------
Cons:
--------------------------
1. The paper claims that using self processing in the core is beneficial, especially by allowing individual feature maps “work” on themselves before being combined with feature maps of different resolution. It would be beneficial to see more explanation on the intuition behind this. Specifically, why is it better to let the feature map work on itself at the core stage as opposed to letting it develop better at the backbone stage, using a heavier backbone?

2. Why does the residual branch as seen in Figure 5 have C1-C3-C1 and not C1-Cx-C1, where x could be 5 or 7 for example? A larger convolutional kernel covers a larger area of the activation map, has a larger receptive field, and may result in better detection performance (although more expensive) when objects of different size (especially larger) are present in the dataset.

3. The top-down and bottom-up blocks as shown in Figure 6 have a specific and distinct structure. What is the reason behind this? Why doesn’t the top-down block have C1-C3-C1-I as opposed to just C1-I?

4. It seems that varying L also varies the total number of self-processing blocks in the TPN core. That is, if L=3 and B=2 (per L), the total number of self-processing blocks in TPN is 2*3=6. However, varying B doesn’t affect the overall number of communication based blocks. Is there a way of varying L independent of B? If not, it is worth mentioning it in the paper for the benefit of the reader.

5. Although different values of B and L result in similar performance, the paper provides empirical evidence that using a larger value of L compared to B yields marginally better performance, in some cases. Even though pointing this out is helpful, the results do not demonstrate that the AP will increase as L increases (while decreasing B to maintain similar cost), e.g., B=2, L=3 (36.7M parameters) yields 41.8 AP, while decreasing B to 1 and increasing L to 5 also yields the same AP, although this setting has 0.4M more parameters. Moreover the paper states the following - “It appears that having more TPN core layers L is slightly more beneficial than having more bottleneck layers B under similar computational budgets. Especially on large objects we observe noticeable differences ranging from 52.5 to 53.7 AP_L”. This statement may imply that increasing L while keeping the cost similar (by decreasing B) yields better performance. However AP_L decreases from 53.7 to 53.4 when L is increased from 3 to 5 (and B decreases from 2 to 1), contradicting this notion. Please provide additional experimental evidence (try out other L>B values) to support the claim above in quotes. Specifically, given similar cost, would increasing L (and decreasing B) always result in better performance? Or are there desirable values of L>B for which the performance is optimal and further increasing L does not help?

6. Paper claims that using more computation at the TPN core results in better performance compared to using a heavier backbone and reiterates the importance of processing at the core level in object detection networks many times. An attempt is made to empirically demonstrate this by showing that ResNet-50+TPN outperforms ResNet-101+FPN, under similar computational cost. However, this result is not sufficient to verify the claim above. It would be interesting to see how ResNet-101+TPN  performs compared to ResNet-50+TPN, under a similar computation budget. Would it be the case that ResNet-50+TPN outperforms ResNet-101+TPN? If so, the above statement claiming that more computation at the core yields better performance can be verified convincingly.

7. Table 2 in page 8 has TPN in the last row, it is not clear what the B and L values are for this configuration. Upon referral to Table 1, it seems like B=2 and L=3. It would be beneficial to mention the B and L values of TPN in Table 2.


**Summary Of The Paper:**

Typically, object detection CNNs consist of a backbone for feature extraction, a core for feature refinement, and a head for predictions. The core is usually a simple arrangement of top-down and/or bottom-up convolutional blocks, used for effectively combining feature maps at different resolutions and this kind of processing is referred to as communication-based processing. This paper claims that the detection performance can be enhanced by the use of a more complex core that enables additional self-processing of individual feature maps (by running them through a series of convolutional layers)  before combining them with feature maps of different resolutions. The paper proposes a new core architecture in object detection CNNs and names it the Trident Pyramid Network (TPN).  In addition, the paper claims that given an additional computational budget, it is better to invest it in a heavier core rather than a heavier backbone, contrary to the mainstream approach. The proposed TPN has two tunable parameters B and L, that control the amount of self processing (B) and the amount of communication-based processing (L).
Experimental studies on the COCO object detection benchmark show that the proposed approach with ResNet-50 backbone and TPN core outperforms existing methods with ResNet-50/101 backbone with FPN or PANet as their cores, under similar computational cost.


**Summary Of The Review:**

The paper is well-written, easy to follow, and the literature review is adequate. The proposed method is well-motivated in the introduction.  Figures describing the proposed method help in better visualization.  Of the two main claims of the paper, one is empirically well-supported, while the other lacks convincing evidence (see main review above). Experimental studies compare the proposed TPN to existing baselines like PANet and FPN, among others, on the COCO dataset. Additionally, the paper  demonstrates empirically that TPN underperforms compared to state-of-the-art transformer based architectures like DETR and Deformable DETR, and in future work, the authors intend to incorporate transformer based processing in the proposed TPN core to further enhance detection performance. The paper may be improved if the cons (see above) are addressed.

---

> ### Author Response · Authors · 2021-11-18
> **Reply to R3 (1/2)**
>
> First of all, thank you for your thorough review and for putting effort in writing down the pros of the method. It's greatly appreciated!
> ***
>
> > **R3.1.1**: *The paper claims that using self processing in the core is beneficial, especially by allowing individual feature maps "work" on themselves before being combined with feature maps of different resolution. It would be beneficial to see more explanation on the intuition behind this.*
>
> **A3.1.1**: We believe that when having multiple communication-based operations in a row, communication tends to saturate (everyone is up to date) and hence becomes superfluous. We argue it is therefore more effective to alternate communication-based processing with sufficient self-processing, such that feature maps have the time to come up with new findings to be communicated.
> ***
>
> > **R3.1.2**: *Specifically, why is it better to let the feature map work on itself at the core stage as opposed to letting it develop better at the backbone stage, using a heavier backbone?*
>
> **A3.1.2**: We believe it is important for the different feature maps (yielding concepts at different scales) to communicate between each other on a regular basis. The backbone however has barely any communication between feature maps, if any. More specifically, the backbone operates by performing computation on a feature map at one scale, resulting in let's say feature map A. Then the backbone downsamples this feature map, and again does some computation on this new map, resulting in feature map B, and so on. At no point in the backbone does B communicate again with A. This is a pity as B might have obtained new insights, while A is left in the dark. To minimize this effect, it is better to use a smaller backbone with a larger core, such that A can quickly catch up with B, rather than using a large backbone with a small core.
> ***
>
> > **R3.2**: *Why does the residual branch as seen in Figure 5 have C1-C3-C1 and not C1-Cx-C1, where x could be 5 or 7 for example? A larger convolutional kernel covers a larger area of the activation map, has a larger receptive field, and may result in better detection performance (although more expensive) when objects of different size (especially larger) are present in the dataset.*
>
> **A3.2**: The residual branch of our bottleneck layers is based on the bottleneck layers from ResNet modules, which use a sequence of C1-C3-C1 convolutional kernels. We left this unchanged. Using a larger kernel will improve the performance, but also (as you mention) increase the computational cost. We do not think it will provide a good trade-off (see for example [1]).
>
> [1] He, Tong, et al. "Bag of tricks for image classification with convolutional neural networks." Proceedings of the IEEE/CVF Conference on Computer Vision and Pattern Recognition. 2019.
> ***
>
> > **R3.3**: *The top-down and bottom-up blocks as shown in Figure 6 have a specific and distinct structure. What is the reason behind this? Why doesn't the top-down block have C1-C3-C1-I as opposed to just C1-I?*
>
> **A3.3**: For the bottom-up operation, a feature map gets updated by a downsampled higher resolution feature map. As we would like all the features from the higher resolution map to contribute to the update, we use a C1-C3-C1 block where the C3 operation with stride 2 guarantees that all features contribute, while downsampling at the same time.
>
> For the top-down operation, a feature map gets updated by an upsampled lower resolution feature map. As in this case all the features from the lower resolution feature map automatically contribute to the update, a simple C1-I block is used instead.
> ***
>
> > **R3.4**: *It seems that varying L also varies the total number of self-processing blocks in the TPN core. That is, if L=3 and B=2 (per L), the total number of self-processing blocks in TPN is 2\*3=6. However, varying B doesn’t affect the overall number of communication based blocks. Is there a way of varying L independent of B? If not, it is worth mentioning it in the paper for the benefit of the reader.*
>
> **A3.4**: The amount of self-processing in TPN indeed does depend on both the number of TPN layers $L$ and the number of bottleneck layers per self-processing node $B$. We will clarify this in our revision. Thanks for pointing this out.

---

> ### Author Response · Authors · 2021-11-18
> **Reply to R3 (2/2)**
>
> > **R3.5**: *Please provide additional experimental evidence (try out other L>B values) to support the claim above in quotes. Specifically, given similar cost, would increasing L (and decreasing B) always result in better performance? Or are there desirable values of L>B for which the performance is optimal and further increasing L does not help?*
>
> **A3.5**: Following the suggestion from R4.2, we trained $15$ different $(L, B)$ TPN configurations of various computation budgets for $12$ epochs using the 1x training schedule. The results are found in Figure 8 of the revised paper. We conclude from the experiments that balanced $(L, B)$ configurations with $L\geq2$ and $B\geq2$ give a better accuracy vs. efficiency trade-off compared to unbalanced $(L, 1)$ and $(1, B)$ configurations. Especially good performance is obtained across various computation budgets when using TPN configurations with $L=2$. See Subsection 4.3 of the revised paper for a more extensive discussion.
> ***
>
> > **R3.6**: *It would be interesting to see how ResNet-101+TPN performs compared to ResNet-50+TPN, under a similar computation budget. Would it be the case that ResNet-50+TPN outperforms ResNet-101+TPN? If so, the above statement claiming that more computation at the core yields better performance can be verified convincingly.*
>
> **A3.6**: Following your suggestion, we trained a ResNet-101+TPN model with $B=2$ and $L=1$, and compared it with a ResNet-50+TPN model with $B=2$ and $L=3$. The ResNet-101+TPN model has a bigger backbone, but a smaller core with $2$ TPN layers less. The results are found in the table below. We can see from the table that the ResNet-50+TPN model outperforms the ResNet-101+TPN model by $0.9$ AP. Additionally, the ResNet-50+TPN model contains $15.0$ M fewer parameters and is $0.7$ FPS faster at inference, compared to the ResNet-101+TPN model (the other computational metrics are similar). This experiment further proves that is more beneficial to add extra computation in the core rather than in the backbone. Thanks for your great suggestion!
>
> | Back | Core | B | L | C |   AP   | AP@50 | AP@75 | AP@S | AP@M | AP@L |  Params  |  tFPS |    tMem   |  iFPS |    iMem   |
> |:----:|:----:|:---:|:---:|:---:|:------:|:---------:|:---------:|:------:|:------:|:------:|:--------:|:-----:|:---------:|:-----:|:---------:|
> |  R50 |  TPN | $2$ | $3$ | $1$ | $41.8$ |   $61.1$  |   $44.4$  | $26.2$ | $46.1$ | $53.7$ | $36.7$ M | $1.6$ | $3.27$ GB | $5.3$ | $0.50$ GB |
> | R101 |  TPN | $2$ | $1$ | $1$ | $40.9$ |   $61.0$  |   $44.2$  | $25.0$ | $45.3$ | $52.6$ | $51.7$ M | $1.6$ | $3.20$ GB | $4.6$ | $0.55$ GB |
> ***
>
> > **R3.7**: *Table 2 in page 8 has TPN in the last row, it is not clear what the B and L values are for this configuration. Upon referral to Table 1, it seems like B=2 and L=3. It would be beneficial to mention the B and L values of TPN in Table 2.*
>
> **A3.7**: The TPN in Table 2 indeed refers to the TPN with $B=2$ and $L=3$. We will clarify this in the revision. Thanks for pointing this out.

---

> > ### Comment · Reviewer_YsZ7 · 2021-11-22
> > **Reply to author rebuttal**
> >
> > Authors please make sure you have addressed A3.7. All others concerns are sufficiently addressed. I recommend acceptance.

---

> > > ### Author Response · Authors · 2021-11-22
> > > **Reply to R3**
> > >
> > > The revised paper is now available, where we have addressed A3.7.

---

### Official Review · Reviewer_EG97 · 2021-11-02

**Correctness:** 4
**Technical Novelty And Significance:** 2
**Empirical Novelty And Significance:** 2
**Recommendation:** 3
**Confidence:** 5

**Details Of Ethics Concerns:**

N.A.

**Main Review:**

Pros:

The results are reasonable, and the method is easy to follow.

Cons:

- First, the existing FPN-based detectors (RetinaNet, etc.) have self-processing module, where a set of conv layers will be applied in $P_{l}$ and $P_{l+1}$ before summing up, and the new aggregated feature map $P_{l}$ will also be processed by a conv layer before output. In other words, the claim that the original FPN has no self-processing module is not correct, and the proposed module is simply to replace the vanilla conv layer with residual block, which is too incremental and heuristic.

- Second, it is a common sense that the FPN module is specifically designed for multiscale representations which is extremely suitable for object detection, and thus it's not surprising that enhancing the representation of FPN outperforms heavier backbone with the same computation budget. This topic has been widely studied in the past few years and the similar idea of the paper (stacking FPN, etc.) has already been proposed by the existing frameworks (e.g., BiFPN and PANet), which leads to limited novelty.

- Third, though it's not an academic publication, the idea of dividing the detection framework into three modules (backbone, core and head) has already proposed in the implementation of mmdetection [1] (backbone, neck and head), which is a widely used platform for object detection.

- Finally, what’s the performance if the proposed TPN added in large backbone, e.g, X-101-DCNv2? To validate the generalization ability of the detector, experiments on large backbones are required.

[1] Chen K, Wang J, Pang J, et al. MMDetection: Open mmlab detection toolbox and benchmark[J]. arXiv preprint arXiv:1906.07155, 2019.


**Summary Of The Paper:**

This paper disentangles the typical detection algorithms into three modules: backbone, core and head, and it proposes a new TPN module to enhance the representation of core module. More specifically, multiple core modules are stacked, each of which is equipped with a residual block for self-processing.

**Summary Of The Review:**

The limited novelty is my main concern to accept this paper, and the module design is too heuristic. I recommend "Reject" as the pre-rebuttal score.

---

> ### Author Response · Authors · 2021-11-18
> **Reply to R2**
>
> > **R2.1.1**: *First, the existing FPN-based detectors (RetinaNet, etc.) have self-processing module, where a set of conv layers will be applied in and before summing up, and the new aggregated feature map will also be processed by a conv layer before output. In other words, the claim that the original FPN has no self-processing module is not correct, ...*
>
> **A2.1.1**: The convolution layers before summing up (*i.e.* before the top-down operation) are linear projection layers meant to transition from backbone feature sizes to core feature sizes. This is a one-time operation and hence cannot be part of a core layer to be stacked. We therefore *conceptually* moved these input projection layers from the original FPN to the backbone, such that the resulting modified FPN layer could be stacked.
>
> The convolution layers after summing up (*i.e.* after the top-down operation) are proper convolution operations with kernel size 3, outputting features of the same size as the input feature size. These operations can and will hence be part of the FPN layer to be stacked. As you mention, the FPN layer indeed contains self-processing. We will modify this in our revision when stated otherwise. Thanks for pointing this out.
> ***
>
> > **R2.1.2**: *..., the proposed module is simply to replace the vanilla conv layer with residual block, which is too incremental and heuristic.*
>
> **A2.1.2**: Our novelty mainly lies in the *abstract* design of the TPN layer, consisting of the sequence of top-down, self-processing, bottom-up and self-processing operations (see also general comment). The bottleneck layer in itself, that we used as self-processing operation for our TPN implementation, cannot be regarded as novelty given it is a layer used by ResNet modules. The introduction of bottleneck layers specifically in the core, could be regarded as a novelty, albeit (as you mention) a minor one.
> ***
>
> > **R2.2**: *This topic has been widely studied in the past few years and the similar idea of the paper (stacking FPN, etc.) has already been proposed by the existing frameworks (e.g., BiFPN and PANet), which leads to limited novelty.*
>
> **A2.2**: See our general comment.
> ***
>
> > **R2.3**: *... the idea of dividing the detection framework into three modules (backbone, core and head) has already proposed ...*
>
> **A2.3**: See our general comment.
> ***
>
> > **R2.4**: *Finally, what’s the performance if the proposed TPN added in large backbone, e.g, X-101-DCNv2? To validate the generalization ability of the detector, experiments on large backbones are required.*
>
> **A2.4**: Following your suggestion, we trained two models using the ResNeXt-101-32x4d-DCNv2 backbone, where we added the TPN core with $B=2$ and $L=3$ to the first model and the BiFPN core to the second model. Given the limited amount of high-memory GPUs in our cluster, we trained the models using a single GPU only (still with batch size $2$), and trained for only $12$ epochs using the 1x training schedule. The results are found in the table below. We can see that the TPN core outperforms the BiFPN core by $0.7$ AP, while having a similar computation budget. Note that these computational characteristics have been obtained as explained in the paper (see caption Table 1), except that we use a batch of two $600 \times 600$ images (instead of two $800 \times 800$ images) to avoid an out-of-memory (OOM) error on the used GeForce GTX 1660 Ti GPU. These new experimental results hence show that the superiority of the TPN core w.r.t. the BiFPN core generalizes to larger backbones. Thanks for your suggestion!
>
> |  Backbone  |  Core | B | L | C | Epochs |   AP   | AP@50 | AP@75 | AP@S | AP@M | AP@L |  Params  |  tFPS |    tMem   |  iFPS |    iMem   |
> |:----------:|:-----:|:---:|:---:|:---:|:------:|:------:|:---------:|:---------:|:------:|:------:|:------:|:--------:|:-----:|:---------:|:-----:|:---------:|
> | X101-DCNv2 |  TPN  | $2$ | $3$ | $1$ |  $12$  | $44.1$ |   $64.9$  |   $47.7$  | $27.9$ | $48.6$ | $58.0$ | $59.1$ M | $1.1$ | $3.60$ GB | $4.7$ | $0.39$ GB |
> | X101-DCNv2 | BiFPN | $-$ | $7$ | $1$ |  $12$  | $43.4$ |   $64.1$  |   $46.9$  | $26.6$ | $47.9$ | $56.7$ | $57.1$ M | $1.2$ | $3.59$ GB | $5.1$ | $0.38$ GB |

---

> > ### Comment · Reviewer_EG97 · 2021-11-29
> > **Post-Rebuttal Score**
> >
> > First thanks for the rebuttal made by the author, and I have carefully read all responses and the comments of other reviewers. Unfortunately, I will not change my score as the following reasons:
> >
> > - The limited novelty is still my main concern to accept the paper. As I claimed in the pre-rebuttal comments, the structure of the (backbone, core (neck), head) has already been proposed and widely used, and the general design of the core (top-down, self-processing, bottom-up and self-processing) is also used in the existing framework (e.g., PANet, with 3x3 conv after top-down, and 3x3 conv after bottom-up). The only difference is the subtle detail of the core implementation is different (e.g., how many layers we stack, replacing the 3x3 conv with multiple residual block). No any new modules are proposed, and the motivation of modification is also intuitive and unclear. In this case, I do not credit the "main contribution claimed in the paper" as one contribution. I give one potential improvement of the paper. If the author can find the limitations of principle in current core design, and propose a new module to address these limitations, the impact of the paper will be significantly improved compared with the current version.
> > - The experiments are still not convincing. I know it's not the fault of  the author since they only have two 6G RAM GPU for experiments, however, the default settings of the large models are significantly reduced to fit the RAM, which leads to less convincing results. And in Table 3 of the appendix, the improvement in large backbone has significantly reduced (1.5%->0.7%). I suspect if the model is trained in default setting, the gap will become even smaller.
> > - Finally, an important issue in the paper I have not found during pre-rebuttal.  All the results reported in the paper are only evaluated in val set and no test-dev results are reported. In principle it's not correct. However, as the generalization of MSCOCO is good, I believe the test-dev results should be very close to the val set, but these results are required.

---

> > > ### Author Response · Authors · 2021-11-29
> > > **Reply to R2**
> > >
> > > Thanks for going over the rebuttal. We would like to make following clarifications.
> > >
> > > > The limited novelty is still my main concern to accept the paper. ... In this case, I do not credit the "main contribution claimed in the paper" as one contribution.
> > >
> > > Our paper contains *three* distinct contributions (see our general comment). We do not consider the contribution you are referring to (or any of the contributions) as the *main* contribution. Take for example a look at the title of the work that states *The importance of processing at the feature pyramid level for better object detection*, clearly referring to a different contribution. We do not believe solely focusing on one aspect of the paper, does the paper justice.
> > >
> > > > No any new modules are proposed ...
> > >
> > > The base modules (convolutions, bottleneck layers, etc.) from which TPN is built. are not new. The TPN module as a whole, with its unique $(L, B)$  configuration, however is new.
> > >
> > > > I give one potential improvement of the paper. If the author can find the limitations of principle in current core design, and propose a new module to address these limitations, the impact of the paper will be significantly improved compared with the current version.
> > >
> > > That's exactly what we did. We discovered that current core designs lack self-processing, and subsequently addressed this limitation by allowing multiple residual blocks $B$ (per self-processing node). Current core designs (such as *e.g.* PANet) namely have configurations of the form $(L, B=1)$ meaning that there is only 1 self-processing operation (*e.g.* convolution) in between each top-down and bottom-up operation. Yet we show in Subsection 4.3 of the revised paper that it is more efficient to have multiple self-processing operations in between each top-down and bottom-up operation ($B > 1$), something which existing core designs cannot do. See Subsection 4.3 of the revised paper for more details.
> > >
> > > > ... they only have two 6G RAM GPU for experiments ...
> > >
> > > We did train on two GPUs from our cluster having at least $10.5$ GB RAM GPU. The GeForce GTX 1660 Ti GPU is only used to get the training and inference FPS of the models.
> > >
> > > > ... the default settings of the large models are significantly reduced to fit the RAM ...
> > >
> > > It is unclear to us which settings you are referring to. We did not change any settings to reduce the memory consumption.
> > >
> > > >  All the results reported in the paper are only evaluated in val set and no test-dev results are reported.
> > >
> > > This is common practice in the field (see for example the DETR paper [1] where they also only use the validation set for their object detection experiments). We could have easily added results on the test set if these were requested earlier.
> > >
> > > [1] Carion, Nicolas, et al. "End-to-end object detection with transformers." European Conference on Computer Vision. Springer, Cham, 2020.

---

> > > > ### Comment · Reviewer_EG97 · 2021-11-30
> > > > **A Quick Response**
> > > >
> > > > Some important comments in the post-rebuttal stage:
> > > > - I have carefully read the paper and comments again and I still have a deep concern of the novelty of the paper. The proposed paradigm has already been proposed in PANet, especially the self-processing module, where the PANet applies **single** 3x3 conv and the TPN uses **$B$** stacked residual blocks. In addition to increase the number of conv blocks and replace them with widely-used residual blocks, i do not see any new concepts, new modules or clear motivation proposed here to address the limitation of current single 3x3 conv for self-process (e.g, limited and fixed receptive fields). To be honest, I feel the proposed concepts (the number of stacked layers, the number of blocks for self-processing) are more close to hyperparameters to calibrate the existing detection paradigm.
> > > > - The author should clearly clarify the specific GPU types of training and testing in the paper, since currently GeForce GTX 1660 Ti with 6G RAM is the only GPU type in the paper , which makes me take it for grant that all the experiments are conduct based on it.
> > > > - The experiments settings of X-101-DCNv2 are different with the default settings (Table 3 in appendix, where the input size is reduced from 800 to 600 to avoid out-of-memory issue). In the original paper and their official implementation[1], the model is trained and tested with 800x1333 input resolution. Reducing the input resolution from 800 to 600 will significantly reduce the performance (1.4%~1.8% from the original RetinaNet paper). Since the improvement of the proposed TPN is also limited (0.7%), I suspect the default settings may further weaken the impact of proposed modules.
> > > >
> > > > Finally I'd like to claim more about the issue of evaluation of the paper. It's definitely NOT the common practice in any machine learning domain that the test set results can be omitted. The annotation of validation set is available and it is used to tune the hyperparameters (e.g., $B$ and $L$ in the paper), then to validate the generalization of the tuned hyperparameters, the test set is used to evaluate whose annotation is not available. Without the test evaluation step, the experiments are not complete. It's true that for some problems (few-shot learning, etc.), only the val is used for evaluation since there is no such evaluation server, however, for generic detection, the test results should not be omitted. Although I do not think it will challenge the effectiveness of the proposed module in MSCOCO set (according to my personal experience), it's a fundamental paradigm to develop any machine learning algorithms and the test results from evaluation server should definitely be added. I will not take it as the reason for rejection since I have not found it in my original review.
> > > >
> > > > [1] https://github.com/facebookresearch/Detectron

---

> > > > > ### Author Response · Authors · 2021-11-30
> > > > > **Reply to R2**
> > > > >
> > > > > > The experiments settings of X-101-DCNv2 are different with the default settings (Table 3 in appendix, where the input size is reduced from 800 to 600 to avoid out-of-memory issue). In the original paper and their official implementation[1], the model is trained and tested with 800x1333 input resolution. Reducing the input resolution from 800 to 600 will significantly reduce the performance (1.4%~1.8% from the original RetinaNet paper).
> > > > >
> > > > > We use a fixed testing procedure on a fixed machine (a machine with a GeForce GTX 1660 Ti GPU) to obtain the **computational characteristics** of models (*i.e.* the training/inference FPS and memory consumption). This is not to be confused with the actual training of models, where we use the multi-scale training scheme from DETR [1] and where we train on bigger GPUs from our cluster.
> > > > >
> > > > > Our default testing procedure (to obtain the computational characteristics) used a batch of two $800 \times 800$ images as input. However, this was not possible for the models with the X-101-DCNv2 backbone due to an out-of-memory (OOM) error. We therefore changed this procedure to use a batch of two $600 \times 600$ images instead. This only changes the interpretation of the FPS and memory consumption metrics in Table 3 (compared to those in Table 1 for example), but in no way influences the performance (*e.g.* the AP) of the models.
> > > > >
> > > > > [1] Carion, Nicolas, et al. "End-to-end object detection with transformers." European Conference on Computer Vision. Springer, Cham, 2020.

---

> > > > > > ### Comment · Reviewer_EG97 · 2021-11-30
> > > > > > **The statement of changing input scale has no impact on detector performance is not correct**
> > > > > >
> > > > > > "We therefore changed this procedure to use a batch of two $600 \times 600$  images instead. This only changes the interpretation of the FPS and memory consumption metrics in Table 3 (compared to those in Table 1 for example), but in no way influences the performance (e.g. the AP) of the models."
> > > > > >
> > > > > > This statement is not correct. It's a common sense that the input scale has significant impact on the performance of the detectors. Take RetinaNet, the base model of the proposed TPN, as one example, if we change the input scale from 600 to 800 pixels for inference, the performance drops from 37.8 to 36.0 on R-101, and drops from 35.7 to 34.3 on R-50 (Table 1e and Figure 2 in [1]). In this case, the default input scale of most detectors on COCO dataset is set as 800x1333 for inference.
> > > > > >
> > > > > > The reason why I require the performance of the proposed TPN on large backbone with full training and inference settings is that, when the proposed module is incorporated in large backbone, even with 600x600 input size, the improvement significantly reduced (1.5% on R-50, and only 0.7% on X-101-DCNv2). Therefore, the improvement may potentially further reduce if evaluated with the default 800x1333 setting, which further weakens the contribution of the proposed modules.
> > > > > >
> > > > > > Furthermore, in Table 1, the author has already determined the hyperparameters of TPN ($B=2$, $L=3$ and $C=1$) which shares similar computation cost as the baseline BiFPN and is also used in Table 3. Therefore, in Table 3, I do not think it is necessary to further conduct experiments on GeForce GTX 1660 Ti. The usage of Table 3 is not to compete with Table 1, but to compare the two X-101-DCNv2 based models with the same settings.
> > > > > >
> > > > > > [1] Lin T Y, Goyal P, Girshick R, et al. Focal loss for dense object detection[C]//Proceedings of the IEEE international conference on computer vision. 2017: 2980-2988.

---

> > > > > > > ### Author Response · Authors · 2021-11-30
> > > > > > > **Reply to R2**
> > > > > > >
> > > > > > > Let's clarify. We only use the two $600 \times 600$ images for the computational characteristics, hence the columns 'Params', 'tFPS', 'tMem', 'iFPS', 'iMem' in the tables.  The APs mentioned in the tables (performance metrics) are computed using the standard procedure where we rescale the images to a shorter edge of $800$ (and maximum of $1333$ for the larger edge). Our statement is hence correct.
> > > > > > >
> > > > > > > Finally, we would also like to reiterate that the reviewer refuses to recognize the three *distinct* contributions of the paper. We get it that the reviewer does not believe our TPN design is novel, despite our many attempts, which we respect. Yet the reviewer fails to discuss or even mention the important observation from the paper, that it is more efficient to perform additional computation at the core level, rather than at the backbone level. Take for example a look at the title that reads: *The importance of processing at the feature pyramid level for better object detection*. We hence believe the review is incomplete.

---

> > > > > > > > ### Comment · Reviewer_EG97 · 2021-11-30
> > > > > > > > **Final Score**
> > > > > > > >
> > > > > > > > I thank the efforts made by the author, and I have carefully read all the comments and list my main concern to accept this paper. Finally my post-rebuttal score is 'Rejection'.

---

### Official Review · Reviewer_NXJS · 2021-11-03

**Correctness:** 3
**Technical Novelty And Significance:** 2
**Empirical Novelty And Significance:** 3
**Recommendation:** 6
**Confidence:** 4

**Main Review:**

In this paper, the authors point out the importance of the core network (the feature aggregation part between the backbone network and the task-specific head) and make a clear point that more computations should be allocated to the core network. The authors also provide a concrete implementation of the core network, called TPN, which performs "self-processing" along with "communication-based processing" during the aggregation of feature pyramids. TPN is compared with the SOTA on the COCO object detection benchmark, and the results support the main claim of the paper.

Weakness:
1. There are many new notions in the paper, including the "core" network, "communication-based processing," and "self-processing." Some statements are obscure, e.g. "too much communication can lead to the core going in circles ... within the core communication-based processing should be alternated with some healthy portion of self-processing." Authors are advised to express their meanings using common notions that are more accessible by researchers in the field.
2. The evaluation is not comprehensive. The backbone-core(neck)-head structure can also be used for semantic segmentation, which is one of the three important tasks in image domain (the classification task is mainly about backbone, and not quite relevant to this architecture). It is a pity that the authors do not validate their design in the segmentation task, leaving the generalization capability of TPN in the fog.
3. It is well-received that consecutive bottleneck layers are added to most feature layers and between top-down and bottom-up aggregation (Fig.4). However, it is not clear why the top-down and bottom-up operations require "self-processing" too (Fig.6). This reviewer does not find any ablation study about whether (and to what extent) this complicated fusion structure outperforms the original simple TD and BU modules, given that "self-processing" modules have already been inserted between TD's and BU's.


**Summary Of The Paper:**

This paper presents a new network architecture for fusing pyramid features in deep neural networks for visual object detection. The authors divide an object detection network into a backbone, a core, and a head. The proposed trident pyramid network, or TPN for short, is a core network structure. The main idea in TPN is that we shall do feature processing along with aggregation in the core part of a network. Experiments are carried out to justify this design.

**Summary Of The Review:**

This paper provides some insights to the design of a three-part network for fine-grained vision tasks. A concrete design is presented and performance gains are demonstrated in the object detection task. However, the experiments are not comprehensive, lacking the evaluation on a relevant task and necessary ablation. The writing of the paper also has room for improvement.

---

> ### Author Response · Authors · 2021-11-18
> **Reply to R1**
>
> > **R1.1**: *Some statements are obscure, e.g. "too much communication can lead to the core going in circles ... within the core communication-based processing should be alternated with some healthy portion of self-processing."*
>
> **A1.1**: Thanks for pointing this out. We tried to convey our intuition, but we will pay attention to this when preparing our revision.
> ***
>
> > **R1.2**: *It is a pity that the authors do not validate their design in the segmentation task, leaving the generalization capability of TPN in the fog.*
>
> **A1.2**: This is a great suggestion. We plan to test this in the near future, but unfortunately we will not get this done by the end of the rebuttal period.
> ***
>
>  > **R1.3**: *However, it is not clear why the top-down and bottom-up operations require "self-processing" too (Fig.6). This reviewer does not find any ablation study about whether (and to what extent) this complicated fusion structure outperforms the original simple TD and BU modules, given that "self-processing" modules have already been inserted between TD's and BU's.*
>
> **A1.3**: It is correct that the used top-down and bottom-up operations also use convolutions (we don't call these *self-processing* as the output feature map will not update the input feature map). The reason to add these is twofold. First, the features of different feature maps are not (perfectly) aligned, and these convolutions help to transition from one feature space to the other. Secondly, the convolutions make the expectation of the residual features zero at initialization (see also Subsection 3.3 of our paper).
>
> It is not clear to us which operations you refer to when talking about '*the original simple TD and BU modules*'. Do you mean TD and BU operations simply consisting of upsampling and downsampling respectively?

---

### Author Response · Authors · 2021-11-18
**General comment**

First, we thank the reviewers for their detailed questions, comments and suggestions. In what follows, we refer to Reviewer NXJS as R1, Reviewer EG97 as R2, Reviewer YsZ7 as R3, and Reviewer MSxk as R4 (based on the top-to-bottom order from OpenReview.net). Comment Y from RX will be referred to as RX.Y.

Here, we address the main comment raised by reviewers R2 and R4 concerning the novelty of the work. Answers to individual reviewers are addressed in separate threads.

### **Where is the novelty?**
R2 and R4, both giving a lower score, question the novelty of the work. We can distinguish three novel items:
1. Technical novelty related to the TPN design.
2. Empirical novelty related to the superior performance of the TPN module compared to existing modules.
3. Empirical novelty regarding the observation that it is more beneficial to perform additional computation in the core rather than in the backbone.

**First**, concerning the TPN design, R2.3 correctly assesses that dividing the detection framework into three modules (backbone, core/neck, head) is not novel. R2.2 and R4.1 correctly assess that stacking Pyramid Networks (PNs) is not novel. Then, where does the TPN design novelty lie?

The TPN design novelty lies in *what* is being stacked, namely the TPN layer. A TPN layer is an abstract module consisting of a sequence of top-down, self-processing, bottom-up and self-processing operations. The module is *abstract* in the sense that it is independent of the precise implementation of the top-down, self-processing and bottom-up operations.

This modular approach allows the user to focus on a specific operation within TPN, such as *e.g.* the top-down operation. When going over the EfficientDet paper [1], a seminal work in the design of stackable PNs proposing BiFPN, we observed that existing PNs paid little attention to the used self-processing operations. Using our abstract TPN template, we designed our own TPN implementation by replacing the self-processing nodes with $B$ consecutive bottleneck layers, the same self-processing operations as used in the popular ResNet backbones.

Our TPN implementation has the unique property that it can balance the amount of communication-based processing and self-processing, by respectively varying the number of TPN layers $L$ and the number of bottleneck layers per self-processing node $B$. This is not possible with existing PNs such as BiFPN. In Subsection 4.3 of the revised paper, we empirically show that $(L, 1)$ configurations as used in BiFPN are inferior to $(L, B)$ configurations with $L\geq2$ and $B\geq2$ under similar computation budgets.

Note that existing PN layers such as the PANet layer [2], match our TPN template. We do not believe this contradicts our novelty claim as our TPN template is more general and allows for many different types of PN layers other than the PANet layer.

**TLDR**: We do not claim to make a major technical contribution with regards to the TPN design. We believe however that the clear separation of operations within the abstract TPN layer, provides an ideal template for future research in PN design. Our TPN implementation, with its unique property of balancing communication-based processing and self-processing, is a first example hereof.

**Secondly**, we show superior performance when comparing our TPN implementation with other PNs such as BiFPN. We believe this empirical result to be of great interest to practitioners in the field.

**Thirdly**, we provide empirical evidence that it is more beneficial to add extra computation in the core, rather than in the backbone. We believe this is an important new insight, which could have far-reaching consequences. Future backbone designs might for example, based on these findings, decide to incorporate more communication-based processing. (See also our response to R3.1.2.)

#### **References**
[1] Tan, Mingxing, Ruoming Pang, and Quoc V. Le. "EfficientDet: Scalable and efficient object detection." Proceedings of the IEEE/CVF conference on computer vision and pattern recognition. 2020.

[2] Liu, Shu, et al. "Path aggregation network for instance segmentation." Proceedings of the IEEE conference on computer vision and pattern recognition. 2018.

---

### Author Response · Authors · 2021-11-18
**New experimental results**

In this comment, we keep track of updates related to new experimental results.

**Update 1**: We have added the results where we compare a ResNet-101+TPN model with a ResNet-50+TPN model of similar computation budget. The experiment further proves that it is more beneficial to add extra computation in the core rather than in the backbone. See R3.6.

**Update 2**: We have added the results where we compare a ResNeXt-101-DCNv2+TPN model with a ResNeXt-101-DCNv2+BiFPN model of similar computation budget. The experiment shows that the superiority of the TPN core w.r.t. the BiFPN core generalizes to larger backbones. See R2.4.

**Update 3**: The revised paper is now available. It contains the many improvements and new experimental results as discussed in the rebuttal. Among these, the revision contains the new experimental results comparing $15$ different $(L, B)$ TPN configurations of various computation budgets. See R3.5, R4.2 and Subsection 4.3 of the revised paper.

---

### Decision · Program_Chairs · 2022-01-20

**Decision:**

Reject

**Comment:**

The submission receives mixed ratings initially. Three reviewers are on the borderline and one reviewer EG97 leans negatively. The raised issues mainly reside on the technical contribution, technical correctness, and experimental validation. In the rebuttal, the authors have tried to address the raised issues and discussed them in-depth with reviewers. However, the discussion does not change the reviewer's mind. After checking all the reviews, rebuttals, and discussions. The AC stands for the reviewer side that the technical contribution is a major issue that ought to be solved. The proposed TPN comes from the summarization of the existing FPN based structure and there are not sufficient insights to make significant improvements. Besides, there are still unsolved issues regarding the technical presentation and experimental validations. The authors are suggested to improve the current manuscript based on these reviews and welcome to submit for the next venue.